

# Similar seed dispersal systems by local frugivorous birds in native and alien plant species in a coastal seawall forest

Bin Liu[1,2], Guohai Wang[1], Yuting An[2], Dandan Xue[2], Libo Wang[2] and Changhu Lu[1]

[1] College of Biology and the Environment, Nanjing Forestry University, Nanjing, Jiangsu, China
[2] Management Bureau of Dafeng Milu National Nature Reserve, Yancheng, Jiangsu, China

## ABSTRACT

Frugivorous birds play an important role in seed dispersal. Alien plant species' seeds are dispersed by local birds in order to establish populations in new habitats. Alien plant species that produce fruits similar to that of native species have the potential to attract local birds, creating new mutualistic systems that are similar to the local ones. In autumn 2018 and 2019, we studied the seed dispersal systems of an alien plant species, *Phytolacca americana*, and a native species, *Cayratia japonica*, in a coastal seawall forest. Both plant species' fruit, frugivorous bird foraging behaviors, seed germination rates, and seedling microhabitats were examined to determine whether the alien species had a similar seed dispersal system to that of the native species. Our results showed that *P. americana* and *C. japonica* had similar fruit type, color, and ripening period. There was a positive correlation between the percentage rate of fruit ripening and the percentage rate of fruit missing for both plant species, indicating that local frugivorous birds have the potential to sufficiently disperse the alien seeds to enable its spread in the coastal seawall forest (simple linear regression, *P. americana*: $\beta = 0.863 \pm 0.017$, $R^2_{adj} = 0.978$, $P < 0.01$; *C. japonica*: $\beta = 0.787 \pm 0.034$, $R^2_{adj} = 0.898$, $P < 0.01$). Eleven bird species consumed the fruits of the alien species or native species during the study period. Similar results were shown across alien and native species in bird foraging behavior (feeding frequency, feeding duration and first stop distance) indicating that a similar seed dispersal relationship had been established between local frugivorous and both plant species. The alien plant had a higher number of fruits carried by birds, suggesting that *P. americana* had a slightly higher fruit consumption than that of *C. japonica* (*t*-test, $P < 0.01$). Alien plant seedlings grow more abundant in forest gap microhabitat (*t*-test, $P < 0.01$). Our results confirmed that bird digestion promotes seed germination success in both plant species. Our study suggests that in a narrow coastal seawall forest, alien plant species can successfully establish their populations by relying on similar seed dispersal systems as the local species.

# INTRODUCTION

Seed dispersal is critical for the maintenance of plant populations (*Butler et al., 2007*; *Wilson & Downs, 2012*) and alien species invasions (*Schierenbeck, 2004*; *Buckley et al., 2006*). Certain

Corresponding author
Changhu Lu, luchanghu@njfu.com.cn

plant species rely on wind, water, or other abiotic forces to spread their seeds far away, while others depend on frugivorous birds for seed dispersal (*Moles et al., 2005*; *Prado, 2013*). Over many years of evolution, a reciprocal relationship between plants and frugivorous birds has been established (*Lu, 2003*; *Li et al., 2019*) in which plants provide necessary nutrients for birds and in return, birds disperse seeds to other habitats and improve plant regeneration (*Camargo et al., 2019*; *Morán-López et al., 2020*; *Galindo-González, Guevara & Sosa, 2000*). Birds are among the most common seed disperser worldwide, and bird-mediated seed dispersal is one of the most important ecological processes (*Saavedra et al., 2014*; *Wang & Ives, 2017*).

Many plant species, including invasive alien plants (*Thabethe et al., 2015*), rely on frugivorous birds for seed dispersal (*Schleuning et al., 2011*; *Duan, Goodale & Quan, 2013*; *Lovas-Kiss et al., 2020*). It is crucial for these alien plant species to establish an adaptation mechanism in order to settle down, build populations, and spread in their new habitat. If alien plants with fleshy fruits do not establish effective dispersal interactions with frugivores present in the local area (*Drummond, 2005*; *Kollmann et al., 2007*), they will fail to settle in a new environment or will have greatly reduced potential for spreading and out-competing native plants (*Aslan, 2011*).

Many studies have looked at the relationship between alien plants and the local birds that feed on their fruits and disperse their seeds (*Renne et al., 2002*; *Gosper, Stansbury & Vivian-Smith, 2005*). Most alien plant species have different traits when compared to those of native species. For example, a study in a coastal belt forest in South Africa showed that plants effectively dispersed by birds were small-seeded, open-habitat species with longer fruiting periods (*Bitani et al., 2020*). In northern California, a study on the proportion of fruits removed by birds showed that all three non-native species, whose fruits differed greatly with native plants in both type and appearance, were assimilated into local bird diets and their dispersal by birds was not limited (*Aslan, 2011*). A study on secondary forest sites on Bonin Island, Japan showed that introduced plant fruits were dispersed differently by native and non-native bird species, and new mutualistic relationships involving native and introduced birds and plants were established in the seed-dispersal system (*Kawakami, Mizusawa & Higuchi, 2009*). Whether seed dispersal systems established by alien plant species are different compared to those of native species has not been determined.

*Phytolacca americana*, an alien fleshy-fruited plant species, and *Cayratia japonica*, a native climbing fleshy-fruited plant, coexist in a coastal seawall forest in the Yellow Sea wetland, Jiangsu Province, China. The forest is a long corridor with planted trees that can provide migration channels, foraging places, and breeding sites for different kinds of birds (*Liu et al., 2020*). *Phytolacca americana* seed dispersal has been reported in both China and its native land (*McDonnell et al., 1984*; *Li et al., 2017*), but *C. japonica* seed dispersal has not been reported. In previous observations of the coastal seawall forest, fruits of both plant species were consumed in autumn by groups of frugivorous birds. Therefore, this forest is an excellent site to study the seed dispersal systems of native and introduced plant species by local frugivorous birds.

We predict that the alien plant species *P. americana* may establish a mutual relationship with the local birds in seed dispersal. Since both *C. japonica* and *P. americana* were the

dominant fruit plants in autumn in the study site, we selected them to examine the seed dispersal systems between the alien plant species and local birds. We compared the two plants' seed dispersal systems according to their fruit characteristics, bird foraging behaviors, seedling microhabitat growth, and seed germination rates.

## MATERIALS & METHODS

### Ethics statement
Field studies were conducted with permission from the Management Bureau of Dafeng Milu National Nature Reserve.

### Study site
The field work was conducted in the Dongchuan coastal seawall forest in Dafeng Milu National Nature Reserve, Jiangsu Province, China (32°59′N∼33°03′N, 120°47′E∼120°53′E). The forest was planted in the 1990s, and was mainly used for windbreak and dike consolidation. The region is characterized by a transition climate between subtropical and warm temperate. The mean annual precipitation is approximately 1,068 mm (concentrated in summer) and the mean annual temperature is 14.1 °C. The vegetation is dominated by *Robinia pseudoacacia*, with smaller areas of *Metasequoia glyptostroboides* and *Populus canadensis*. The width of the forest is about 20∼50 m. One side of the forest is an intertidal zone with the main vegetation types *Spartina alterniflora*, *Suaeda glauca*, *Phragmites communis,* and *Imperata cylindrica*, and the other side is farmland and pond culture. Annual typhoons blow over the trees, exposing many open forest gaps. Many bird species live in the forest, including the light-vented bulbul (*Pycnonotus sinensis*), common blackbird (*Turdus merula*), and black-billed magpie (*Pica pica*). From April to May and September to October, many passerines, mainly warblers and flycatchers, travel through the forest in order to complete their migration life cycle. In autumn, this area also has a small number of other fruiting, such as *Lycium barbarum*, *Sapium sebiferum*, *Ligustrum lucidum*, and *Paederia scandens*.

### Field observation
#### *Fruit and seedling investigations*
From September to October 2018, we randomly chose 20 focal plants from each target species along the coastal seawall forest. The fruit type, size, shape, and color, ripening period, and height were recorded across 20 sampled individuals from each species. The fruit's size (mm) was defined as its diameter measured by a caliper ($n = 20$ for each plant species). The fruit's color was the color of peel. The ripening period, dates of the natural phenology of the plants, was from September to October. The maximum fruit's height was defined by measuring the highest fruits and taking the average value from the 20 sampled individuals. To calculate the rates of ripe and missing fruits, we counted the number of ripe fruits from five randomly sampled individuals across each species every 5 days. The number of fruits was the total fruits on one sampled tree. The missing fruits was the difference between the number of total fruits and the number of total fruits 5 days later. Seedlings were found within plots ($r = 25$ m, $n = 15$ for each plant species) around the individual

selected plants. Seedling microhabitats were inspected at the same time: the forest gap was formed after trees were fallen by wind and there was enough sunshine for seedlings, and the understory was where the sunlight could hardly reach the ground. Tree density was measured by sampling method. 20 plots ($10 \times 10$ m$^2$) were randomly selected along the forest. Each species' plant density (the number of plants divided by the area of the plots) was investigated in the study site.

### Bird feeding behaviors

We observed the behaviors of birds feeding on the two plant species' fruit in the forest between September and October 2018 and 2019. The focal individual sampling method was used to quantify visitation and fruit removal by birds at every sampled plant individual. 20 focal individuals of each plant species were investigated. Since most bird foraging occurred at dusk and dawn, observations were carried out from 6 am to 9 am, or 4 pm to 6 pm on sunny days. The total observation time was 85 h. Each observation continued until the birds had disappeared. We counted the number of fruits swallowed and pecked by each bird species and the total foraging time in every observation. The focal bird's first stopping distance (the distance between the first resting place of the visiting birds and the sample plant, also regarded as the potential minimum seed dispersal distance) was measured using a laser range finder. We calculated the total number of feeding visits by each bird on each plant species across the study period. The average feeding duration (in seconds) on each plant species was also recorded.

### Seed germination trials

We compared the cumulative germination success of defecated seeds by local frugivorous birds with that of hand-cleaned seeds and intact fruits. All seeds and fruits were collected from the study site in October 2018. Seed germination experiments were carried out from June 20 to August 20, 2019. Defecated seeds from each individual bird, hand-cleaned seeds, and the intact fruits were planted in separate bowls with a diameter of 10 cm. Each experimental group contained 100 seeds, planted equally across 20 bowls. All bowls were then placed in a laboratory balcony and watered daily during the first month. Seeds were considered germinated when seedlings emerged through the soil surface. The number of seeds germinated was recorded at least once every 15 days between early June and mid-August. Once counted, seedlings were removed from the tray.

## Data analysis

We compared the maximum fruit's height and fruit size of each plant species using a $t$-test. A $t$-test was also used to compare the density of the seedling microhabitat growth. We used simple linear regression to test the correlation between each species' fruit ripening rate and missing rate. We used a $t$- test to compare the total feeding frequency of each bird species on each plant species, the average feeding duration (in seconds), the amount of fruit consumed per visit, and the first stopping distance of the focal bird. If the data distribution was not normal before or after square root treatment, we used the $U$-test. ANOVA tests were used to compare the germination success of defecated seeds to hand-cleaned seeds and intact fruits. After that, the LSD test was used to compare the difference between each
**Table 1  Fruit characteristics and plant density of the two plant species in the study site.** Number in brackets indicates the sample size. Mean and SD are displayed in the table.

| | *Cayratia japonica* | *Phytolacca americana* |
|---|---|---|
| Fruit type | Berry | Berry |
| Fruit size (mm) | 10.0 ± 0.9 (20) | 7.9 ± 0.7 (20) |
| Fruit shape and color | Round, black | Round, Purple black |
| Fruit ripening period | August to October | August to October |
| Fruit height (m) | 0.2 ± 0.1 (20) | 1.3 ± 0.3 (20) |
| Plant density (individuals/m²) | 0.5 ± 0.3 (20) | 0.1 ± 0.1 (20) |

other. All analyses were performed using SPSS 26.0 and the software package R version 3.4.4 (*R Core Team, 2018*) was used to make the plots. The level of statistical significance was set at $P < 0.05$.

# RESULTS

## Fruit characteristics of plant species

In autumn, *P. americana* and *C. japonica* are the most common fruiting species in the coastal seawall forest. Both fruit types are berry. When ripe, their colors change to black or purple-black. There was a significant difference in the fruit size between the species ($t$-test, $t = 8.252$, $df = 38$, $P < 0.01$) (Table 1). *Cayratia japonica* has bigger fruits than *P. americana*.

Phytolacca americana mainly grew in the forest gap habitat, while *C. japonica* was widely distributed in the understory along the seawall forest. The fruit ripening period of the two plant species basically overlapped, and both ripened between August and October. There was a significant difference in the height of fruit distribution between the two species ($U$-test, $Z = -5.420$, $P < 0.05$, $n = 40$). Fruit height of *P. americana* is higher than *C. japonica*.

## Fruit feeding and seed dispersal by frugivorous birds

Ripe fruits were constantly eaten by frugivores, and no other animal groups were observed to eat *P. americana* or *C. japonica* fruits during the daytime. The relationship between the percentage of fully-ripe fruits and the percentage of fruits taken by birds showed a significant positive correlation for both plant species (simple linear regression, *P. americana*: $\beta = 0.863 \pm 0.017$, $R^2_{adj} = 0.978$, $P < 0.01$; *C. japonica*: $\beta = 0.787 \pm 0.034$, $R^2_{adj} = 0.898$, $P < 0.01$) (Fig. 1). This suggested that the more ripen fruits, the greater proportion of missing fruits were eaten by birds or fallen on the ground.

A total of 11 bird species were observed feeding on the fruits. Vinous-throated parrotbill (*Paradoxornis webbianus*), blue-and-white flycatcher (*Cyanoptila cyanomelana*), and grey-backed thrush (*Turdus hortulorum*) were the three species that only fed on *Phytolacca americana*. Black-napped oriole (*Oriolus chinensis*) was the species that only fed on *C. japonica*. The rest of the bird species fed on both fruits of the two plants (Fig. 2).

The *Pycnonotus sinensis* and *Pica pica* had the highest feeding frequencies on *P. americana* and *C. japonica*, respectively. *Pycnonotus sinensis* and *Sturnus cineraceus* had the longest

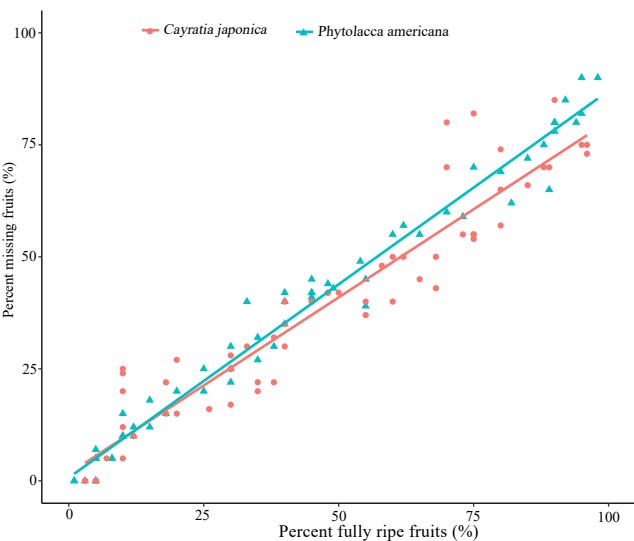

**Figure 1** **Relationship between percent fully ripen fruits and percent of fruit missing caused by bird visiting.** Percent missing fruits are based on the remaining fruits of each sampled plants ($n$ =10). Simple linear regression is used for statistical analysis.

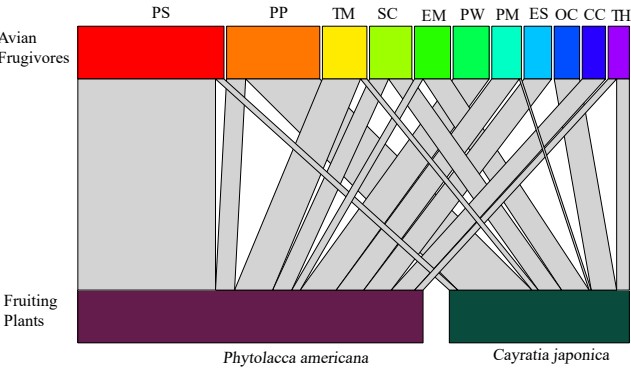

**Figure 2** **Correspondence relationship between avian frugivores and fruiting plants based on feeding behaviors.** Widths of connecting lines denote the number of observed interactions (wider represents higher intensity of visiting). Avian frugivores: PS: *Pycnonotus sinensis*; PP: *Pica pica*; TM: *Turdus merula*; SC: *Sturnus cineraceus*; EM: *Eophona migratoria*; PW: *Paradoxornis webbianus*; PM: *Parus major*; ES: *Emberiza spodocephala*; OC: *Oriolus chinensis*; CC: *Cyanoptila cyanomelana*; TH: *Turdus hortulorum* .

foraging times on *Phytolacca americana* and *Cayratia japonica* fruits, respectively. The *Sturnus cineraceus* was the species with the furthest first stopping distance after feeding on both plants (Table 2). However, there were no significant differences in bird feeding frequency ($U$-test, $Z = -1.116$, $P > 0.05$, $n = 18$), feeding duration ($t$-test, $t = 1.960$, $df = 16$, $P > 0.05$), or first stopping distance ($t$-test, $t = 1.401$, $df = 16$, $P > 0.05$) between *P. americana* and *C. japonica*. However, the number of *P. americana* fruits consumed per visit was significantly larger than the number of *C. japonica* fruits consumed ($t$-test, $t = -3.941$, $df = 16$, $P < 0.01$) (Table 2).

Liu et al. (2021), *PeerJ*, DOI 10.7717/peerj.11672

**Table 2  Bird feeding behaviors on the native and alien plant species.** Number in brackets indicates the sample size. Mean and SD are displayed in the table, except for feeding frequency which represents the total number of visits.

| Bird species | Feeding frequency /times | | Feeding time/s | | Number of consumed fruits per visit | | First stopping distance/m | | Feeding pattern | Seasonal type |
|---|---|---|---|---|---|---|---|---|---|---|
| | CY | PA | CY | PA | CY | PA | CY | PA | | |
| *Pycnonotus sinensis* | 4.0 | 65.0 | 10.0 ± 7.1 (4) | 14.2 ± 7.7 (65) | 1.5 ± 0.6 (4) | 4.1 ± 1.9 (65) | 9.3 ± 7.2 (4) | 13.4 ± 7.7 (65) | S | R |
| *Turdus merula* | 3.0 | 18.0 | 6.7 ± 3.5 (3) | 7.3 ± 3.2 (18) | 1.3 ± 0.6 (3) | 3.7 ± 2.4 (18) | 25.0 ± 5.0 (3) | 28.6 ± 14.6 (18) | S | R |
| *Cyanoptila cyanomelana* | 0 | 11.0 | 0 | 5.2 ± 3.5 (11) | 0 | 1.8 ± 1.2 (11) | 0 | 8.3 ± 5.3 (11) | S | T |
| *Sturnus cineraceus* | 11.0 | 9.0 | 26.9 ± 17.6 (11) | 10.1 ± 5.3 (9) | 1.9 ± 0.8 (11) | 4.6 ± 2.1 (9) | 75.0 ± 47.0 (11) | 70.0 ± 26.0 (9) | S | R |
| *Pica pica* | 35.0 | 9.0 | 16.7 ± 10.6 (35) | 10.0 ± 6.5 (9) | 2.1 ± 1.2 (35) | 4.2 ± 2.8 (9) | 29.9 ± 16.1 (35) | 23.3 ± 7.1 (9) | S | R |
| *Oriolus chinensis* | 12.0 | 0 | 11.9 ± 8.1 (12) | 0 | 2.6 ± 1.1 (12) | 0 | 31.9 ± 25.1 (12) | 0 | S | T |
| *Turdus hortulorum* | 6.0 | 4.0 | 14.5 ± 6.3 (6) | 7.8 ± 5.7 (4) | 1.8 ± 0.8 (6) | 2.8 ± 2.2 (4) | 42.5 ± 16.7 (6) | 19.3 ± 10.3 (4) | S | T |
| *Paradoxornis webbianus* | 0 | 17.0 | 0 | 5.8 ± 2.9 (17) | 0 | 3.6 ± 1.8 (17) | 0 | 6.7 ± 2.3 (17) | S/P | R |
| *Parus major* | 1.0 | 13.0 | 5.0 (1) | 6.8 ± 3.4 (13) | 1.0 (1) | 2.2 ± 0.9 (13) | 12.0 (1) | 12.2 ± 4.3 (13) | S/P | R |
| *Eophona migratoria* | 13.0 | 4.0 | 20.8 ± 13.7 (13) | 10.0 ± 4.8 (4) | 1.7 ± 0.9 (13) | 4.5 ± 3.1 (4) | 26.0 ± 12.4 (13) | 15.0 ± 5.8 (4) | S/P | R |
| *Emberiza spodocephala* | 0 | 13.0 | 0 | 11.6 ± 3.6 (13) | 0 | 2.0 ± 0.9 (13) | 0 | 7.2 ± 4.6 (13) | S/P | W |

**Notes.**

CY, *Cayratia japonica*; PA, *Phytolacca americana*.

Feeding pattern: S, Swallow; P, Pecking.

Seasonal type: R, Resident birds; T, Travel birds; W, Winter birds.

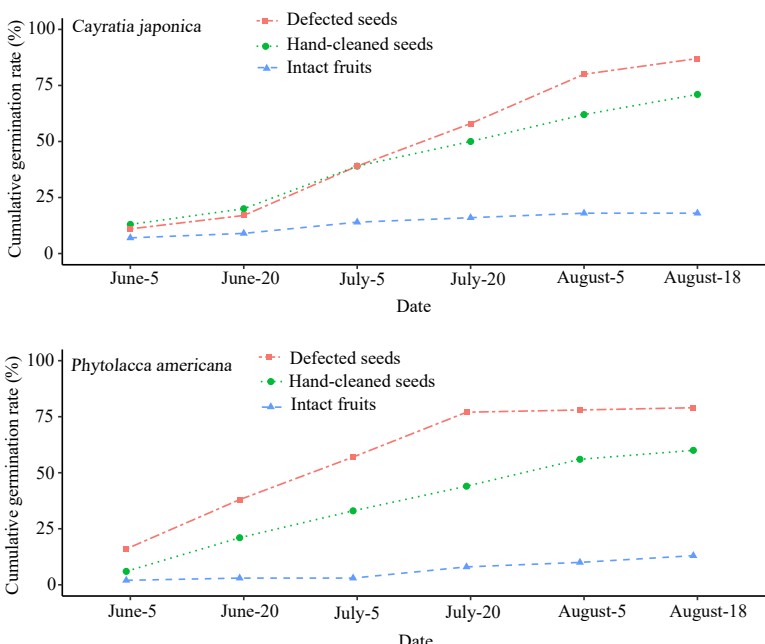

**Figure 3  Cumulative seed germination rates of the two plant species in three different groups.** Each group contains 100 seeds.

## Seed germination rate

The seed germination rate of intact fruits was lower than both hand-cleaned seeds and defecated seeds from both plant species. *Phytolacca americana*'s final seed germination rates across the three different groups were as follows: 79% for defecated seeds, 60% for hand-cleaned seeds, and 13% for intact fruits (Fig. 3). Additionally, *C. japonica*'s final seed germination rates were 87%, 71%, and 18%, respectively (Fig. 3). There was a significant difference of seed germination rates across the different groups between *P. americana* (ANOVA: $F_{2,15} = 13.283$, $P < 0.01$) and *C. japonica* (ANOVA: $F_{2,15} = 4.591$, $P < 0.05$). The seed germination of intact *P. americana* fruits was significantly lower compared to hand-cleaned seeds (LSD-test, $P < 0.01$) and defecated seeds (LSD-test, $P < 0.01$). The same result was shown in *C. japonica*. This showed that fruit pulp removal by local frugivorous birds' digestion or other methods could promote seed germination success across both plant species.

## Seedling microhabitats

We randomly found 300 *P. americana* seedlings and 280 *C. japonica* seedlings along the seawall forest in the study site. There were differences in microhabitat characteristics between the two plant species. *Phytolacca americana* seedlings preferred the forest gap habitat, and significantly more seedlings were found in the forest gap habitat than in the understory habitat (t-test, $t = 5.684$, $df = 28$, $P < 0.01$) (Fig. 4). However, *C. japonica* seedlings did not show an obvious preference when selecting a microhabitat (t-test, $t = 0.092$, $df = 28$, $P > 0.05$) (Fig. 4).

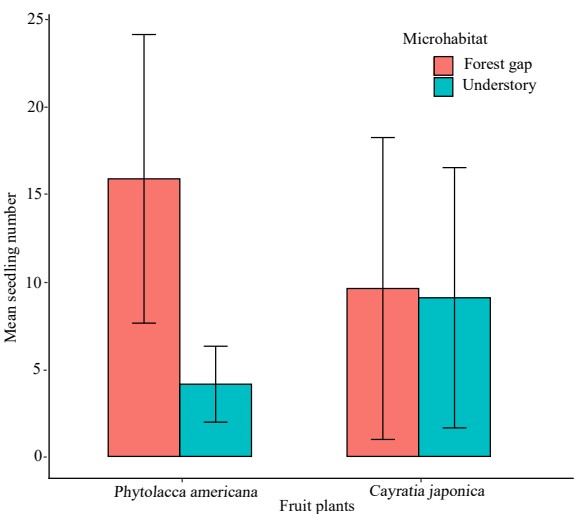

**Figure 4** **The mean seedling number of the native and alien plant species growing in two different microhabitats.** Thirty plots ($r = 25$ m) are investigated. Error bars show the standard deviation.

## DISCUSSION

### Similar fruit characteristics between the two plant species

Bird-mediated seed dispersal is important for the establishment of many fleshy-fruited plant species (*Wenny, 2001*). Many alien plants with fleshy fruits are able to establish effective dispersal interactions with local birds based on having attractive fruit characteristics that are similar to those of native plants (*Gosper, Stansbury & Vivian-Smith, 2005*). Our study showed that the alien *P. americana* species had similar characteristics (e.g., fruit type, shape, color when ripe, and ripening period) to the native *C. japonica* species (Table 1). This similarity suggests that the alien species *P. americana* has great potential to have its seeds dispersed by local frugivorous birds that have already established a positive relationship with the native plant species *C. japonica*.

Previous studies have found that small fruits and seeds are more effectively removed and dispersed than larger ones (*Rey et al., 1997*), and birds prefer visiting trees with larger crop sizes (*Sallabanks, 1993*; *Korine, Kalko & Herre, 2000*). A study in a coastal belt forest in South Africa found that fleshy-fruited invasion plant species effectively dispersed by birds were small-seeded, open-habitat species with longer fruiting lengths compared to native plant species (*Bitani et al., 2020*). If the former conclusion is true, and based on the fact that *P. americana* has smaller fruit size but more seeds than *C. japonica* (*McDonnell et al., 1984*; *Hansen & Goertzen, 2006*), *P. americana* could use these traits as an advantage over *C. japonica* in order to improve its dispersion efficiency.

The forest gap microhabitat formed by light disturbance was more suitable for the growth of alien plant species (*Willamson & Fitter, 1996*; *Ridenour & Callaway, 2001*). We found that alien plant species seedlings tended to occupy the forest gap microhabitat, which was more suitable for seedling growth. However, the native plant seedlings showed no preference in microhabitat selection. This suggested that alien plant species seedlings

had developmental potential for successful population regeneration in the coastal seawall forest. These results were consistent with the findings of *Burnham & Lee (2010)* and *Martins et al. (2004)*.

## Similar seed dispersal systems

A fruiting species can be consumed by a variety of potential bird dispersers, and a potential bird disperser can remove a variety of fruit species (*Dugger et al., 2018*; *Schneiberg et al., 2020*). In our study, seven out of 11 local frugivorous birds consumed fruits from both plant species, suggesting that a stable relationship between the alien plant and local birds had already been established. Three bird species were found to only consume the alien species, showing that it had a wider popularity with local bird species (*Whelan & Willson, 1994*; *Levey & Martínez del Rio, 2001*). Ten out of 11 and eight out of 11 bird species who ingested the whole fruit may be potential seed dispersers of the alien species and the native species in study site, respectively. Four bird species were pecking birds since pecking may have damaged the seeds (Table 2).

*Pycnonotus sinensis* is a small and common frugivorous bird species that we studied during our field observation. Because of the lack of insects, ripening fruits were their main food resource in the coastal seawall forest in autumn. We typically observed three to five *Pycnonotus sinensis* individuals foraging on *Phytolacca americana* trees with large crop sizes and purple-black ripe fruits. Ultimately, *Pycnonotus sinensis* interacted the most with the alien species and was its main seed disperser.

Overall, we did not find an obvious pattern of preference across the local birds' selection and utilization of fruits from these two plants, which reflected the degree of similarity between the introduced plant species *P. americana* and the native species *C. japonica*. We found some different feeding behaviors between the two plant species, but according to the number of consumed fruits, *P. americana* showed more easily used by birds. We found that the frugivores feeding on the fruits were mostly resident birds. Migratory birds contributed less to seed dispersal according to the lower feed frequency and shorter feed duration of the three migratory species, *Oriolus chinensis*, *Turdus hortulorum*, and *Cyanoptila cyanomelana*. Therefore, we concluded that resident birds were the main contributors to *P. americana* and *C. japonica* seed dispersal. This may be due to the fact that birds migrating through the seawall forest in autumn were mostly insectivorous flycatchers and warblers (*Liu et al., 2020*).

## Effect of birds on seed germination rate

Bird ingestion may increase the seed germination rates of different plant species, and many fruit plant species, including invasive species, have been shown to benefit from ingestion by frugivorous birds (*Combs et al., 2011*; *Carrion-Tacuri et al., 2012*; *Czarnecka, Orlowski & Karg, 2012*; *Thabethe et al., 2015*).

Our results showed that the seed germination rates from intact fruits of both plant species were lower than from ingested and manually de-pulped seeds (Fig. 4), which was consistent with the results of *Barnea, Yom-Tov & Friedman (1990)*, *Yagihashi, Hayashida & Miyamoto (1999)* and *Traveset (1998)*. This suggested that local frugivorous bird ingestion

similarly and positively affected the germination success of alien and native plant species in the study site. The local birds were legitimate seed dispersers/seed predators or pulp peckers of the fleshy-fruited alien plant. The higher seed germination rate of ingested and manually de-pulped seeds may be attributed to the removal of pulp or appropriate seed coat abrasion, but more trials are necessary to prove this conclusion. The final cumulative germination percentage of the native plant species was similar to that of the alien plant species across three different groups. This suggested that the seed dispersal pattern in *P. americana* was similar to that of *C. japonica*.

Many alien plants have interfered with the dynamic between frugivores and native plants (*Farwig & Berens, 2012*). Alien species with abundant and nutritious fruits may prevent birds from visiting native plants. This reduces seed dispersal success and recruitment (*Hansen & Müller, 2009*; *Mokotjomela, Musil & Esler, 2013*). In our study, the alien plant *P. americana* had more abundant fruits and attracted more visiting frugivores than the native plant *C. japonica*. The mutualistic interaction between the alien plant and native frugivores is likely to lead to an invasion with a negative impact on the local ecological environment. Appropriate measures should be taken to control a potential invasion of *P. americana* in the study site.

## CONCLUSION

The alien species *P. americana* showed similar fruit characteristics to the native plant species *C. japonica*. A mutualistic seed dispersal relationship between the alien plant species and local frugivorous birds, similar to the native one, has been established. Local frugivorous bird digestion could promote the seed germination rate of the alien plant, and its seedling was more likely to occupy the forest gap microhabitat than the native plant seedling. All of these factors have allowed the alien plant species *P. americana* to remain and rapidly expand in Dongchuan coastal seawall forest.

## ACKNOWLEDGEMENTS

We are grateful to Shengbin Xie, Yong Zhang, and Baodong Yuan for their help in data processing. We also thank the staff at the Dafeng Milu National Nature Reserve for their contributions to this field of study.

### Funding

This work was supported by a project funded by the Priority Academic Program Development of Jiangsu Higher Education Institutions (No. PAPD). The funders had no role in study design, data collection and analysis, decision to publish, or preparation of the manuscript.

### Grant Disclosures

The following grant information was disclosed by the authors:

The Priority Academic Program Development of Jiangsu Higher Education Institutions (No. PAPD).

## Competing Interests

The authors declare there are no competing interests.

## Author Contributions

- Bin Liu conceived and designed the experiments, performed the experiments, analyzed the data, prepared figures and/or tables, and approved the final draft.
- Guohai Wang analyzed the data, prepared figures and/or tables, and approved the final draft.
- Yuting An analyzed the data, authored or reviewed drafts of the paper, and approved the final draft.
- Dandan Xue and Libo Wang performed the experiments, authored or reviewed drafts of the paper, and approved the final draft.
- Changhu Lu conceived and designed the experiments, authored or reviewed drafts of the paper, and approved the final draft.

## Animal Ethics

The following information was supplied relating to ethical approvals (i.e., approving body and any reference numbers):

The Administrative Bureau of Dafeng Milu National Nature Reserve approved this research.

## Ethics

The following information was supplied relating to ethical approvals (i.e., approving body and any reference numbers):

The Administrative Bureau of Dafeng Milu National Nature Reserve approved the animal ethical inspection.

## Field Study Permissions

The following information was supplied relating to field study approvals (i.e., approving body and any reference numbers):

Field studies were conducted under the permission from the Administrative Bureau of Dafeng Milu National Nature Reserve.

## Data Availability

The raw measurements are available in the Supplemental Files.

## Supplemental Information

Supplemental information for this article can be found online at http://dx.doi.org/10.7717/peerj.11672#supplemental-information.

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
