# Peer review of "Similar seed dispersal systems by local frugivorous birds in native and alien plant species in a coastal seawall forest"

_PeerJ, doi:10.7717/peerj.11672_

## Round 0.1 · original submission · Major Revisions

Dear Dr. Liu,

The manuscript has been reviewed by three reviewers. All of them agree on the value of this study but also that it needs substantial revision. The study could be published in PeerJ if you are able to deal with all suggestions and corrections made by the reviewers.

Cheers,

Marcial

Reviewer 1 ·

Basic reporting

In the present manuscript, Liu et al. compared the bird-seed dispersal of an alien (Phytolacca americana) and a native plant species (Cayratia japonica), two species that produce fleshy fruits with similar morphological traits and on the same season. These fruits were promptly (i.e., as soon as they mature) consumed by 11 bird species. The study concludes that the alien species has no constraints in its seed dispersal by native frugivorous birds, which promotes its establishment and expansion through disturbed microhabitats (e.g., forest clearings) at the study area.
• I find it an interesting study; however, it needs a thorough revision in the English language so that the text can be clearly understood. I suggest corrections for some examples in the section “General comments”.
• The introduction seems to be overall well structured, but I think that there are sections that need a bit of more work, especially between lines 63-75. There are poorly constructed sentences here (lines 63-64 and 74-75) that renders their message hard to convey to the reader.
• The manuscript’s structure conforms to PeerJ standards. The raw data and the permits for the field work were supplied.
• There is a mismatch in the number of figures cited in the main text and at the end of the manuscript (4 vs. 7).
• There is also some missing information regarding the meaning of the values presented in both tables and in the barplots (please see my comments in the section “General comments” for more details).

Experimental design

• This research lies within the scope of the PeerJ journal.
• The research question is present at the end of the introduction section. However, it is buried within an extremely long sentence (lines 85-90), which makes it to be somewhat difficult to understand. I suggest the authors to rewrite this long sentence in a clearer manner and split it into smaller sentences.
• In the methodology section, there are multiple missing details regarding the experimental design that must be provided (lines 118, 120-125, 130-131).

Validity of the findings

• The results of the study seem reasonably interpreted and the conclusions related to the research question. However, it is difficult to critically assess the results without a more detailed description of the sampling design (see my comments in the next section).

Additional comments

Line 26: delete the word “much”. It has a subjective meaning.

Line 27: “…which were attracting the visitation of local birds.”. I don’t see this statement anywhere in the results and discussion sections. Moreover, it was not tested. Please, delete it or develop/justify this idea in the manuscript.

Line 30: At the beginning of the sentence, replace “11” by “Eleven”.

Line 35: Replace “larger amount of the fruits” by “higher number of fruits”.

Line 43: Replace “Seed dispersal was” by “Seed dispersal is”. Here and throughout the manuscript, especially in the introduction section and in the description of the study site, the authors should use the present tense when presenting facts.

Line 63-64: It is not easy to understand what the authors are trying to say in this sentence.

Lines 85-90: The sentence is too long. Consider splitting it.

Line 86: Why did you assess the seed dispersal of only one native species (i.e., Cayratia japonica)? I read in lines 113-114 that there were more fleshy-fruited plant species at the study site. Was it because they were scarce? Because they produce fruits that are morphologically different to those of Phytolacca americana and you only wanted to test fruits with similar morphological traits?

Line 110: It is enough to state only the months when migratory birds pass through the study site. Adding also the season is redundant, so you may delete it.

Line 118: You inspected plants every 50 metres in transects or randomly? How many plants of each species did you sampled?

Line 119: This is the first mention to these variables: “fruit characteristics”, “fruit height”, and “microhabitat”. Therefore, you must state what they stand for. For example, what is the “fruit height” variable? Maximum fruit height on each plant? Average fruit height on each plant? Height of each measured fruit?

Line 120: How many fruits per individual plant were measured?

Line 121: How was the plant density measured? Within random plots? Which area?

Lines 123-124: The number of ripe and missing fruits were assessed in the whole plant or sampled in specific branches? And it was assessed in how many individual plants?

Line 125: What was the sampling design to find those seedlings? Was it within plots around selected individual plants? If that was the case, what was the plot area?

Lines 130-131: How many individual focal plants were sampled?

Line 136: I suggest replacing “feeding frequency number” by “feeding visits” or other straightforward term.

Line 137: Based on Table 2, I presume that the “feeding duration” is the number of seconds that each bird spends on each individual plant. Therefore, I suggest changing the text segment “feeding duration per time” to “feeding duration (in seconds)”.

Line 153: What kind of generalized linear model? The authors should detail what kind of model was used.

Lines 155-156: Apart from “first stopping distance”, the meaning of these variables is not very clear (see my comments above). What is the “feeding amount per time”?

Line 175: Replace “As time going on” by another phrasing. For example, “As the time progresses”.

Line 180: It is insufficient to present only the p values. Please, provide the coefficient estimate and standard error, statistical test, degrees of freedom, and p- value. These parameters should also be presented separately for the two models (one for the alien and another for the native plant species).

Line 182: Replace “11” by “Eleven”.

Lines 194-195: Change “was the species whose first stopping distance longest after” to “was the species with the farthest fist stopping distance after”.

Lines 196-197: Please, present the results of the Mann-Whitney test in the same way it is in line 171. Additionally, for consistency, both the t-value and degrees of freedom should be reported for all t-tests.

Line 202: Replace “300” by “Three-hundred”.

Line 205: Delete “much”. It is a subjective term. Alternatively, you may consider replacing it by “significantly”.

Line 210: Move this sentence to the methods section.

Lines 216-217: Since you found significant differences in the ANOVA tests, why not use post-hoc tests to assess which treatment(s) is/ are leading to such differences? The authors state that seed ingestion by birds promoted seed germination, which is supported by the results. But this benefit might not be significantly different from the hand-cleaned treatment.

Line 249: Replace “7” by “Seven”.

Line 253: There is a mistake in the way these references are presented.

Lines 262-263: The authors found no “(…) obvious “preference” in the selection and utilization of the seeds (…)”. It seems to me that this lack of selection was found because all bird species were analysed as a whole, which might mask the variability in the response of each species. When I look at the data provided in Table 2, I see that most bird species display substantial differences in “feeding frequency” and/or “feeding time” between plant species. This suggests that, although most birds visit both plant species, they still have “preferences” toward either the alien or the native plant. Of course, a formal analysis would be needed to test this assertion.

Line 302: “itis” to “it is”.

Line 303: “nativeone” to “native one”.

Lines 303-305: Cumbersome sentence. Consider rewriting it.

Table 1: What do these values represent? Are they mean ± standard deviation?

Table 2: Are these values representing means? What is the sample size? A value representing the variability in measurements (i.e., standard deviation) should also be included.

Figures: The number of figures cited in the text do not match those found at the end of the manuscript (4 vs. 7 figures, respectively). This is because the plots belonging to the same figure are separated in distinct figures, which is odd.

Figures 4 and 5 (at the end of the manuscript): What do the error bars represent? This must be indicated in the figure caption.

·

Basic reporting

I am attaching the original .pdf document with the corresponding annotations and suggestions that make up most of my review.

The question addressed in this work is interesting and important for me to answer when it comes to understanding how are the dispersal processes by frugivorous animals of invasive species in community and facilitation contexts. In general terms, the work is self-contained and both the data collected and the corresponding analyzes are necessary to address the question posed.

I believe that the level of English must be significantly improved for publication, there are many sentences that must be rewritten and many others shortened.

The general structure seems correct to me, although it is necessary to review the redundant information and delve more deeply into some points of greater interest and scope. Tables and figures are correct, in the .pdf notes I suggest several modifications that I think can improve the figures and tables qualitatively. There is a serious error in the code of the figures that is carried over almost the entire manuscript!

In the discussion, I miss a deeper thinking on the interpretation of the results obtained. Address issues such as the effect of this loss of dispersal events for native species caused by “disperser theft” caused by an invasive species. Also, some examples with other mutualistic or antagonistic interactions (for example, it is common to study the effects of invasive species in pollination or parasitism systems). This type of cases could be useful to understand the role that similarity-dissimilarity of traits has in the adaptation of invasive species in a context of biological interactions.

Experimental design

As in the previous section, specific suggestions and recommendations can be found in the attached document.

I have liked reading the experimental design, although there are several points that should be explained more specifically. It is an ambitious design as it covers quite a few important points in a dispersion system, which is highly valuable.

Practically all the samples are poorly explained in the text (crop size estimations, seedling count, methodology of focal observations, selection and justification of study sites, etc.). I believe that it is necessary to be more specific when explaining how each of these samplings has been carried out and why it has been in each specific way.
The number of fruits used to characterize both species seems little to me and it would be good to be able to measure more fruits and explain how they have been collected.

Clearly detailing the procedures for these types of measures significantly improves attributes such as replicability or repeatability in field studies. If it is a problem of space in the text, I suggest detailing these procedures in Supplementary Material.

Validity of the findings

NO COMENT

Additional comments

I would like to congratulate you. The question addressed seems interesting to me and the way to approach it is correct. I miss a clearer definition of the starting hypotheses and a more in-depth development of the discussion, paying attention to the consequences and natural processes that these results suggest.

·

Basic reporting

The authors explore the hypothesis that the invasive plant Phytolacca americana being similar in fruiting to the native plant Caryatia japonica can take advantage of a pre-existing frugivorous assemblage and so have a successful population establishment.

The idea and objective of the study is very interesting and original. The objective of the study is clear and justified. Also, there is few literature on invasive plants’ seed dispersal, so this study is a valuable contribution.

However, there are important issues that need the attention of the authors to allow a better understanding.

English language needs to be thoroughly revised throughout the whole manuscript. The current state of the manuscript makes reading comprehension difficult and slow. Verbal tenses are often mistakenly used (e.g. the use of past for present events), and there are many parts of the text that need rephrasing for a better understanding. I have corrected the English in the abstract and introduction section (see pdf attached) to give an idea to the authors of the necessary corrections. I am not a native speaker so some of my corrections may not be appropriate. I suggest the authors ask for the help of an English native speaker, or someone with an advanced level of it.

The manuscript would also benefit from some additional references on the topic of invasive fleshy-fruited plant species dispersal and seed dispersal in general (e.g: Traveset, A., & Richardson, D. M. (2014). Mutualistic Interactions and Biological Invasions. Annual Review of Ecology, Evolution, and Systematics, 45(1), 89–113.)

The structure is clear, and the article is well self-contained. Figures and table need a revision, more explanation in their captions, as well as more information the material and methods section (see detailed comments in General comments).

Experimental design

The idea of the study is well defined, and it fills a knowledge gap on invasive plants seed dispersal.

The general hypothesis could be complemented with minor ones, to better support why the specific experiments have been settled. Also, some of the recorded data is not used in the discussion (such as bird feeding behaviour or stopping distance). I believe this data can be very useful to discuss the legitimacy of the different species as dispersers (see in General comments).

Materials and methods section needs more developing. It is not clear how many study sites there were. Also, some experiments are not clear or are not fully supported by a previous hypothesis (such as the seedling growth by microhabitat or the birds stopping distance).

Authors should also include the number of individual plants studied for each experiment. From the metadata I can guess 5 individual plants were marked for the calculation of removal rates, but it is not clear how many individuals were used to calculate bird visiting/feeding rates and how many plants were used to measure the fruits characteristics and fruit height. The choice of different statistical tests would also benefit from additional explanation.

Validity of the findings

All underlying data but plants density have been provided in the metadata. The use of some statistical tests could be better clarified, but all test values are duly reported.

I have one point regarding the conclusions and discussion. Seed dispersal is not explicitly measured in this study, so I advise authors to review all those parts in the manuscript where they talk about it. While several species are acting as dispersers (proven by the faeces you germinated), the bird identity of these faeces is not made explicit. Some of the frugivore species that you documented may be acting as pulp thieves (peckers) or seeds predators (e.g. Eophona migratoria; Shi, Y., Li, X., & Guo, Z. (2012). Tree fruits consumed by Eophona migratoria in winter. Journal of Biology, 29(3), 20-23).

Please be aware of this when taking conclusions or speculating.

In addition, I have some more specific comments that the authors should address. They are detailed in “General comments” sections by lines to facilitate the revision process.

Kind regards,

Elena Quintero

Additional comments

Abstract

Line 23: In the biological cycle of a plant, germination comes first to seedling grow. Therefore, I recommend stating before results on seed germination and latter results on seedling growth based on microhabitats.
Since differential seed deposition is not measured, I wouldn’t consider seedling establishment as a proxy. There are many survival factors that can differentiate proportions of seed deposition and seedling establishment.

Line 35: when you say “carried by birds” you only consider consumed/transported fruits right? I imagine you not accounting for birds pecking on fruits. Bird feeding behaviour is not very much discussed in the manuscript. I think you can use this behaviour data to discuss on the role of birds as legitimate disperser.

Line 36: Careful you did not measure seed dispersal per se, to do that you would need to prove that bird consumers actually act as dispersers (you only did this during the germination experiment with unidentified bird feces). Consider changing “seed dispersal” for “fruit consumption”.

Line 37: “preferred” is a bit confusing. I suggest you use other term as “grow more abundant in”.

Line 38: Start phrase with: “Our results support that...”. No need to use “could” in this phrase, if you have actually proved that seed germination is better with bird ingested seeds, you can speak safely.


Introduction

Most of this section is written in past tense, when events are presently occurring. Please check all verbal tenses throughout the text and change it to present when necessary.

Line 47-49: link both sentence by a comma instead of a point for better comprehension.

Line 50-51: this is a big claim, do you mean “one of the most important processes”? Seed dispersal while being very important process is not the only important one.

Line 55-57: Why does this process need to be rapid? Please provide a reference if leaving this phrase.

Line 57-60: link both sentence by a comma instead of a point for better comprehension.

Line 70-74: this is in contraposition to what you say in line 58, where it only mentions “local” birds. Maybe change to “frugivores present in the local area”.

Line 74-75: Please rephrase this sentence, since it is hard to follow.

Line 78: what do you mean by “artificial” trees?

Line 81: Perhaps you have confused Li et al 2019 reference, with this one:
Li, N., Yang, W., Fang, S., Li, X., Liu, Z., Leng, X., & An, S. (2017). Dispersal of invasive Phytolacca americana seeds by birds in an urban garden in China. Integrative Zoology, 12(1), 26-31. The reference you citate does not study Phytolacca americana.

Line 82: please change feed by consumed.

Line 85: please change dominate by dominant

Line 89: seedlings do not “select” microhabitat purposely, maybe choose a different term that applies better like “thrive”, “grow”,…


Materials & Methods

It may be interesting to mention somewhere that Cayratia japonica is a climbing plant, to better understand its seed dispersal and seedling growth.

Line 108-109: I suggest you write common names with small letters, and Latin names between parenthesis.

Line 119: What do you mean by fruit height? Is it the mean height at which the fruits are located in the plant? Please clarify.

Line 121-123: this is the first notice for the reader of a second study site. How many sites did you study? If there were two study areas (at least for some of the experiments), you should also include it in the materials & methods section.

Line 121-123: these results do not have supporting numbers anywhere or statistical significance. Can you please provide it?

Line 123-124: I don’t follow this experiment. How did you exactly measure the ripening rate? Did you always referenced ripen fruits to the original crop size or did you calculate the percentage based on the remaining fruits? I find confusing the biological meaning of this relationship.

Line 126: why did you only selected two microhabitats? Are these two the most important ones or clearly distinguished? Maybe you had a prior hypothesis based in these 2 microhabitats? Please explain the reader why you took this decision.

Line 130: How many observation hours did you do and how many plants did you observe?

Line 133: As I understood, the focal “subject” in this case is the plant, not the bird. I suggest you remove “focal” in this phrase to avoid confusion. Maybe you mean something like “birds that visited the plant were observed until they were out of sight”?

Line 134: this information is not present in the results neither in the metadata. Behaviour is divided in swallowed or pecked. Did you use ‘dropped fruits’ or ‘taken in flight’ for some experiment? If so, please justify it.

Line 141: How many seeds did you planted for each experimental group? From the metadata I see you used 20 bowls with 5 seeds each, is that correct? Please include it.

Line 143: Do you have the identity of the bird feces? Were they all from the same bird species?

Line 153: Could you please justify why you used a GLM and not a Linear model? was it because the data followed a non-liner distribution?

Line 156: depending on what did you use t-test or U-test? Please clarify


Results

Please check figure numbers and letters, they are mixed up (e.g. figure 1A and 1B = figure 1 and figure 2). Figure captions need more explanation, especially explain what axis represent and error bars.

Line 169: this is very interesting. Did any of the two plants had a longer fruiting period than the other?

Line 171-172: please provide numbers and statistical significance if you have measured this. You may also want to include this in table 1.

Line 175-176: Say explicitly Cayratia turns black and Phytolacca turns purple black for more clarity.

Line 177: no other animal was observed during your focal observations, but you did not monitored plants during the afternoon/night. Please state this only holds true for the time you monitored.

Line 180-181: I am not sure to understand this experiment. How do you make sure missing fruits are consumed by birds and not fallen?

Line 182, 202 and 249: when starting a sentence with a number it should be written it in letters, please adjust this.

Line 182-190: I suggest you shorten this section by stating only the names of those species feeding in one of the two plants. The rest (a total of 7) are common and previously mentioned in M&M and in fig. 3.

Line 194-195: why did you measure stopping distance? Was this a proxy for minimum dispersal distance. This is not explained in the introduction hypothesis.

Line 198: I suggest you change “feeding amount each time” by “number of fruits consumed per visit”.

Line 203: why seedlings are searched around mother trees? Weren’t they searched in microhabitats?

Line 210: move this sentence to Materials and methods section.

Line 212 and line 216: “controls” is confusing, I suggest you change it to “groups”, since the controls would be hand-cleaned seeds and intact fruits (not defecated seeds).


Discussion

Line 227, 288, 300 and 26: English correction: The term much similar in this article is often mistakenly used. “Much similar” seems to refer to the comparison of the 2 plants with another unknown system. I recommend you remove the “much” and leave just “similar” for all instances throughout the text.

Line 227: I recommend changing “indicates” by “suggests”, since you are speculating.

Line 236: can you provide a reference for the fact that Phytolacca contains more seeds? If you also calculated this, you should include it in the M&M and results sections.

Line 237: Seed dispersal effectiveness is not only based in consumption frequency but also in quality factors such as gut treatment, microhabitat deposition or seed predation that are not accounted in this study (see Schupp 1993 Vegetatio https://doi.org/10.1007/978-94-011-1749-4_2.). I advise something more restrained as: Phytolacca can use these traits as an advantage over Cayratia improving the effectiveness of its dispersion.

Line 244: This is very interesting! you can link it to plant colonization processes in forest gaps (especially worrying for invasive species). See: Burnham, K. M., & Lee, T. D. (2010). Canopy gaps facilitate establishment, growth, and reproduction of invasive Frangula alnus in a Tsuga canadensis dominated forest. Biological Invasions, 12(6), 1509-1520.
Or
Martins, S. V., Júnior, R. C., Rodrigues, R. R., & Gandolfi, S. (2004). Colonization of gaps produced by death of bamboo clumps in a semideciduous mesophytic forest in south-eastern Brazil. Plant Ecology, 172(1), 121-131.

Line 251-252: since you have recorded bird feeding behaviour, I suggest you use this information to discuss the potential of species as dispersers (i.e. the 4 species of peckers probably are not dispersing much), and the implications.

Line 261-262: these are already mentioned results and do not need to be repeated unless they are followed by an explanation.

Line 276-279: The seed germination Inhibition hypothesis you mention holds true for many fruits species, not only invasive ones (see: Cipollini, M. L., & Levey, D. J. (1997). Secondary Metabolites Of Fleshy Vertebrate‐Dispersed Fruits: Adaptive Hypotheses And Implications For Seed Dispersal. The American Naturalist, 150(3), 346–372. https://doi.org/10.1086/286069)

Line 284-285: Is not possible to generalize for all species. Some may be seed predators/pulp peckers. Since you have not identified the identity of the bird’s faeces, you should indicate so.


Conclusion

Line 301: I suggest you replace the world “system” by “mutualistic relation”

Line 302 and 303: need space between words “it is” and “nativeone”

Figures

For a better understanding of figure 1 I think it would help plotting fruit removal along *time* for both species, so it shows depletion rates. See for example figure 1 in Cullen, A. C., Farley, K. E., Pagano, S. S., Gallagher, F. J., & Holzapfel, C. (2020). All in the timing: How fruit nutritional content influences the timing of fruit consumption of two invasive shrubs. Plant Ecology, 221(10), 951–963. Yet, I am aware this may not be in the scope of the authors, so just consider it a suggestion.

Figure 3. the interaction links indicates the number of visits or the number fruits consumed? You indicate links are ‘relative’, do you mean relative to plants or to what?
Also, I propose you add a grey colour to the links, to facilitate interpretation, otherwise with too many black lines are a bit messy.

Figure 4. I suppose y-axis is a mean number since it has error bars. How is it calculated? (e.g. per site, per m2…)

Figure 6. y-axis shows *accumulated* seed germination

---

## Round 0.2 · Minor Revisions

The three reviewers have concluded that the manuscript has improved significantly since the last version. But the three reviewers also clearly suggest that there is still room to improve the manuscript.
Please, consider the suggestions made by the three reviewers, and resubmit the manuscript.

Reviewer 1 ·

Basic reporting

The English language was considerably improved in the current version of the manuscript. I would like to congratulate the authors for the good work. The figures are now corrected but there are some typos in the captions (for more details, please see my general comments below).
Although the manuscript was notoriously improved (especially in grammar and the provided methodology), I still have a few comments that are detailed under “General comments”.

Experimental design

no comment

Validity of the findings

no comment

Additional comments

Lines 31-34: This sentence needs to be rewritten (at least lines 31-32; consider dividing it in two sentences) because it is hard to understand. I suggest the authors to rewrite to something such as “There were no significant differences in bird feeding frequency, feeding duration, and first stop distance between the alien and the native plant species. These similar results were shown across …”.

Lines 50-51: “Birds are the most common seed disperser in the world…” Unless you have a reference to support this statement, I suggest rephrasing to “Birds are among the most common seed dispersers worldwide”.

Line 51: Change “meditated” by “mediated”.

Line 62: “different seed dispersal systems” still seems a bit vague to me. Given the references presented afterwards, it seems to me that rephrasing to “Most alien plant species have different traits when compared to …” would fit better and have a more straightforward meaning.

Line 75: State that the native plant species also produces fleshy fruits.

Line 77: Because it starts a sentence, do not abbreviate the genus. Write “Phytolacca americana”.

Line 116: Do not use the word “trees” to refer to the individual plants because they are not trees. Instead, change here and where applicable in the manuscript (lines 119, 120, 124, 174, 175) to, e.g., “20 sampled individuals from each species” or “20 sampled focal plants from each species”.

Line 116: Write the units in which you measured fruit size.

Line 118: In that case, don’t you think that it would make more sense to rename the variable “fruit’s heigh” to “maximum fruit’s height”?

Lines 160-161: The use of post-hoc tests (and which one was used) after ANOVA should be stated here.

Line 162: Please, provide a reference for R software and state that it was used to make the plots.

Line 170: Because it is starting a sentence, the generic epithet in Phytolacca americana should not be abbreviated.

Lines 181-182: Please, provide the regression estimated parameters (i.e., the beta coefficient).

Line 191: For consistency, throughout the manuscript, always abbreviate the generic epithet of the scientific names except when the name is written for the first time or when it is starting a sentence.

Line 236: There is a typo here. Change “Americana” to “americana”.

Lines 255-256: Cumbersome sentence. For example, the part where it is mentioned that those are pecking birds should come at the beginning of the sentence, before saying that pecking may have damaged the seeds.

Line 256-258: Why were they considered seed dispersers? I’m assuming that it is because they ingest the whole fruit, but this information must be stated here.

Line 268: It would be easier to understand this sentence if the authors would replace “feed amounts” by, for example, “number of consumed fruits”.

Line 304: “in autumn in a coastal seawall forest”. This part of the sentence doesn’t add any important information here because the fruits of both plant species are similar irrespective of the season and habitat (right?). Therefore, I suggest the authors to delete this text segment.

Table 2: In the caption, the first sentence doesn’t look right. I suggest the authors to replace it by, e.g., “Bird feeding behaviour on the native and alien plant species.”

Table 2: As you did in the main text, I suggest the authors to replace the name of the variable “feeding amount” by another name that makes it easy to understand what was really measured here.

Figure 1: In the caption, there is a typo. Change “analysi” to “analysis” and delete “(A) Phytolacca americana”.

Figure 2: There is a mistake in the caption.

Figure 3: Replace “Accumulate” by “Cumulative”. Also delete “which planted averagely in 20 bowls.”.

Figure 4: Replace “30” by “Thirty” because it is starting a sentence.

·

Basic reporting

The authors have substantially improved the quality of the manuscript. The improvement in writing and the level of English is evident. The organization and design of the figures and tables has also improved.

In the general comments section some suggestions and necessary changes are detailed, indicating the specific line in each case.

Experimental design

A detailed description of the ratios for ripen fruits and missing fruits is still missing.
Some other suggestions on experimental design are noted in the general comments section.

Validity of the findings

NO COMMENT

Additional comments

Lines 31 -34: The wording of this sentence makes it difficult for me to understand, even though the result is simple and robust. It could perhaps be said: “Similar results were shown across alien and native species in bird foraging behavior (feeding frequency, feeding duration and first stop distance) indicating that a similar seed dispersal relationship had been established between local frugivorous birds and both plant species.

Lines 83-87: At the end of the introduction, perhaps before this paragraph of objectives, it might be good to briefly develop the hypothesis or starting point, what facilitation or competition processes can be found and what consequences they could have. I consider that an explanation of this would improve the quality of the manuscript as it would help the reader to understand why it may be necessary to compare these seed dispersal systems.

Line 109: Confusing. Perhaps “small number of other fleshy fruited plant species such as Lycium..”

Line 114: Change “..we randomly inspected..” for “ we randomly chose 20 focal trees from each target species along the coastal..”

Line 118: Fruit´s height is confusing for me, in Table 1 for Cayratia japonica the average Fruit height is 0.2 (m). How have you measured this trait? Did you measure the distance of the tallest fruits to the ground?

Lines 119-121: Despite the McDonnell reference, it is not clear to me how these ratios are calculated. If values of 100% are reached, I suppose that everything refers to an initial available cropsize. Is that correct? How are the missing fruits calculated? Is it a subtraction or have you based on signals in the Infructescence? These results are key to affirming that there is no limitation in the dispersal of any of the target species, so it is necessary to define well how they have been calculated.

Line 130: “20 focal stands”. In each focal stand could inhabit more than one individual plant. There are stands with both species co-ocurring or are species-specific stands? It is confusing because the incongruence between "focal individual sampling" and "20 focal stands". Be more specific please.

Lines 179-180: Perhaps an example of how these values have been measured in the field, which was measured, and how these ratios have been calculated, can be useful to understand and suggest clarifications to improve the explanation of these results.

Line 212: The selection of the bar.plot of fig. 3 leads to a misunderstanding. What is being compared in the t-tests are the microhabitat preferences within each species. I think the best thing to do would be to represent the seedlings found in each microhabitat of one species next to each other, and in the other part of the graph the other species. Substitute the order of place (Forest gap and understory) for the species, and use the colors for the Microhabitats. It is only a suggestion and there are many possibilities, but what is being compared in the statistical test does not agree with the supporting figure.

Line 213: “We randomly..” In M&M section you explain: Seedlings were found within plots (r=25m, n=30) around the individual selected plants. Maybe just write: "We found 300 P. americana seedlings.."

·

Basic reporting

English language has greatly improve in this new version. The manuscript has a good and clear structure as well as a clearly defined goal. Supporting literature is enough and most data is provided.

Experimental design

The research question is very well supported (i.e. the comparison of dispersal systems for the species). Authors could increase the justification and significance of the manuscript by briefly adding a general hypothesis at the end of the introduction (e.g. invasive plant taking advantage of an already established dispersal system of a native fruiting plant and the implications this may have).

Materials and methods are better explained in this new version. Yet I am still missing a few more details on the section “Fruit and seedling investigations”. It is not fully clear to me how authors performed some experiment (see general comments below).

Table 2 has greatly improved and the merging of figures 1 and 4 looks much better, allowing for comparisons between the native and the invasive species.

Overall, I congratulate the authors for the great improvement of the manuscript and after addressing the issues raised in here, I think it will be ready for publication.

Validity of the findings

No comment

Additional comments

Abstract

Line 17-18: Local birds do not disperse plants with that purpose (establish populations), that is rather the purpose of the plants.

Line 25: I suggest writing the words “types”, “colors” and “periods” in singular.

Lines 31-34: very long phrase, please consider splitting in two. Also significance values are not in context since you are not stating comparisons in the sentence.


Introduction

Line 55: there is a red dot, change to black font.

Lines 62-63: What you say here is in contraposition to what you say in lines 71-73. From what I understand from the text I suppose you mean that, based on the literature available to date, it appears that the dispersal systems of invasive plants differ from those of native plants in the area they colonize. However, I am not certain that this is what the authors mean. Please review these two sentences.

Line 82: change the second ‘and’ for a ‘by’.


Material and Methods

Line 109: add species after ‘fruiting’

Line 116: how many fruits per plant did you take?

Lines 117-118: The ripening period length refers to the dates you have done field work or to the natural phenology of both plants? Please clarify.

Lines 119-121: If you did this experiment as in McDonell et al 1984, please reference it for further support. However McDonell et al. used bagged branches as reference to calculate percentage, but I think you did not. Could you please explain with further detail how you calculated ‘ripe and missing fruits percentage’? To what total number is this calculation referred, to the number of ripen fruits present in the previous revision maybe? Or were the percentages visually estimated?

Line 121: to what does the r in the parenthesis refers to? radius of the plots? Please clarify in the text.

Lines 121-122: As I get from the supplementary data you provide, for the seedling microhabitat experiment you selected 15 plots per species, of which half corresponded to forest gap and the other half to understory, is this right? Please develop a bit more the experiment description.

Line 125: These 20 plots are new to the reader (as far as I understood), therefore please introduce them as such. When you say “of the 20 plots” seems confusing, as if they have been mentioned previously.

Lines 133-134: is this information 'number of fruits per visit' present in the table 1?

Line 139: this reference is out of context here, I suggest deleting it.


Results

Lines 168-169 and lines 172-174: since you only have 2 species, please indicate which has bigger fruits and which is higher.

Lines 174-175: Please revise the English in this sentence. This is the first notice you distinguish between plant’s age (not stated in M&Ms), also in the Table 1 tree density is grouped by species (not divided by age). Did you perform any statistical analysis? Supplementary data for tree density is not provided.

Lines 189-197: Please stick to common or latin names for the birds species, but do not use them interchangeably.

Line 210: Did you test differences in germination between species? In figure 3 seems that Caryatia seedlings germinate in a higher proportion.

Line 213: randomly found? If I understood correctly an active search for seedlings was done in 20 plots (so, not random).


Discussion

Lines 255-256: it is not possible to tell how much each bird dispersed (since this was not actually measured), please use conditional when speculating. Maybe these birds did not even disperse at all.

Lines 256-258: These numbers do not coincide with the previous statements, are you considering all species as potential dispersers (including peckers)? Either change the sentence or move before lines 255-256.

Line 261-263 and 272: choose between using latin or common names for birds.

Line 269: P. americana showed significant higher frugivory, and this may give it an advantage over C. japonica, however but you can’t prove it actually had more effective dispersal. This depends on the different birds feeding on it (some frugivores may be more effective than others, or some may not even disperse) the differential contribution between bird species is key for determining dispersal effectiveness.

Line 290-292: germination difference between species could actually be tested with you data, did you do perform any analysis?

Line 309: I suggest replacing “this coastal..” by “Dongchuan coastal…“


Tables and figures

Table 2: feeding frequency and feeding amount need more explanation. It is not clear if feeding frequency refers to number of fruits per visit, or total number of visits. Then feeding amount is not clear if refers to number of plant individuals visited or number of birds visiting at the same time.

Figure 2: some error in the figure legend, one sentence is duplicated.
Link colours are now blue and black and looks messy. I suggested filling the link with a light grey color in order to differentiate it from the background.
Also if enough space I suggest adding birds latin name abbreviations above the rectangles, so it would facilitate visualisation.

Figure 3. Accumulate needs a “d” in the end

---

## Round 0.3 · accepted · Accept

The authors have successfully answered questions and added suggestions by three expert reviewers. The manuscript has been much improved in two rounds of revision. It is ready now for publication.